# Redox Balance in Cancer in the Context of Tumor Prevention and Treatment

**DOI:** 10.3390/biomedicines13051149

**Published:** 2025-05-09

**Authors:** Paweł Sutkowy, Przemysław Czeleń

**Affiliations:** 1Department of Medical Biology and Biochemistry, Ludwik Rydygier Collegium Medicum in Bydgoszcz, Nicolaus Copernicus University in Torun, Karlowicza 24, 85-092 Bydgoszcz, Poland; 2Department of Physical Chemistry, Faculty of Pharmacy, Collegium Medicum, Nicolaus Copernicus University, Kurpinskiego 5, 85-096 Bydgoszcz, Poland; przemekcz@cm.umk.pl

**Keywords:** oxidants, antioxidants, cancer, prevention, diagnosis, treatment

## Abstract

Malignant neoplasms constitute a substantial health concern for the human population, currently ranking as the second leading cause of mortality worldwide. In 2022, approximately 10 million deaths were attributable to cancer, and projections estimate that this number will rise to 35 million in 2050. Consequently, the development of effective cancer treatments and prevention strategies remains a primary focus of medical research. In this context, the impacts on the redox balance are being considered. The objective of this study was to present the current knowledge on oxidation and reduction processes in cancer. This review discloses the intricate and multifaceted interplay of oxidoreductive systems during carcinogenesis, which engenders discordant findings in the domain of tumor prevention and treatment. This study also examines the controversies surrounding the use of antioxidants, including their impact on other therapeutic interventions. The review offers a comprehensive overview of the existing knowledge on the subject, concluding that personalized and precise anticancer therapies targeting the redox processes can serve as both effective diagnostic and therapeutic tools.

## 1. Introduction

The prevalence of cancer is increasing at an alarming rate, reaching the point where it has become one of the most significant global health concerns. Cancer is currently the second leading cause of mortality in developing countries, with cardiovascular disease representing the most common cause of mortality. The most recent report, prepared by researchers from the International Agency for Research on Cancer (IARC) and the American Cancer Society (ACS), indicates that approximately 20 million new cancer cases and nearly 10 million cancer-related deaths are anticipated in 2022. A demographic-based forecast suggests that the annual incidence of cancer will reach 35 million by 2050, representing a 77% increase from the value in 2022 [1]. The projected increase in the future burden is attributed to the physical and environmental risk factors that have emerged as a consequence of globalization and the socioeconomic transformations that have accompanied it. These include pollution, ultraviolet radiation, the adoption of new lifestyles, and an unhealthy diet (characterized by poor dietary habits and physical inactivity, in addition to smoking, among other factors) [2]. In 2022, lung cancer was the most frequently diagnosed neoplasm, representing almost 2.5 million new cases, or one in eight neoplasms, worldwide (12.4% of all cancers globally). This was followed by cancers of the female breast (11.6%) and colorectum (9.6%). Lung cancer was also the leading cause of cancer death, being responsible for an estimated 1.8 million deaths (18.7%). The next most common causes were colorectal (9.3%) and liver cancers (7.8%) [1].

It is currently accepted that reactive oxygen species (ROS) and reactive nitrogen species (RNS), along with inflammation, represent a key factor in the development of neoplastic cells [3]. Even non-cancerous stimuli that cause chronic inflammation may initiate the process of carcinogenesis. This is often linked to increased production of ROS, not only inside cells of the tissue but also due to immune cells migrating to the site of inflammation [4]. Such effects may culminate in the development of chronic disease, including cancer [3,4]. A concentration of ROS that exceeds the antioxidant capacity results in oxidative stress [5], which may in turn lead to DNA damage, genome instability, the modulation of signaling pathways (especially nuclear factor-κB, NF-κB, and mitogen-activated protein kinase, MAPK), and eventually the promotion of neoplastic transformation [3,4]. Elevated levels of ROS are present in malignant tumors at all stages of carcinogenesis [6]. The formation of reaction products between ROS and biomolecules is implicated in the pathogenesis of cancer and a multitude of other pathological conditions due to their toxicity and/or abnormal structure (e.g., 8-hydroxy-2-deoxyguanosine (8-OHdG), malondialdehyde (MDA), 4-hydroxy-2-nonenal (4-HNE), and carbonylated proteins) [7]. ROS are oxidant molecules that are an integral part of the normal functioning of an organism. In relatively high concentrations, they are involved in maintaining the elevated metabolic activity of a tissue [8]. These active forms of oxygen are also of great importance in the development of cancerous cells and their metastasis [9]. However, as previously stated, when ROS concentrations exceed the antioxidant capabilities of both normal and neoplastic cells, the resulting toxicity causes oxidative damage to biomolecules, leading ultimately to apoptosis or even necrosis and inflammation [8,10]. In other words, ROS play a significant role in the regulation of cellular aging by participating in the control of growth and differentiation of both normal and cancerous cells [11]. For example, it is already well established that a diet high in fat and sugar, which is typical of the Western diet, induces oxidative stress and cellular aging, thereby contributing to the development of age-related cancers [12,13]. At the same time, it has been shown that elevated levels of ROS can induce cancer regression or impede its progression and completely inhibit metastasis [14]. However, when cancer cells adapt to excessive ROS concentrations by improving their antioxidant capacity, they are capable of undergoing metastasis [14]. This suggests that increasing antioxidant capacity (eliminating excessive oxidative stress) in cancer cells is necessary for their development. Furthermore, cancer adaptation to oxidative stress may occur in other ways. It has been demonstrated that oxidative stress resulting from mitochondrial dysfunction can promote cancer cell senescence, thereby inhibiting cancer cell growth [15,16]. Senescent cancer cells are capable of producing a distinctive protein profile, known as the senescence-associated secretory phenotype (SASP), which can influence neighboring cells. While oxidative stress-induced senescent cancer cells undergo atrophy accompanied by inflammation, the activated stromal cells are able to create a new tumor microenvironment [11]. This fact is important in cancer treatment. In addition, this phenomenon can be associated with other unexpected reactions [11]. For example, doxorubicin, a frequently utilized clinical pharmaceutical, has been demonstrated to stimulate the formation of senescent cells and the SASP, which can ultimately result in the development of doxorubicin resistance and cancer progression [17]. The aging of cancer cells due to oxidative stress induced by chemotherapy can result in an improved tumor condition [18]. In contrast, berberine, a naturally occurring alkaloid, augmented the therapeutic efficacy of X-rays in treating liver cancer (decreased proliferation) while also inducing cellular senescence [19]. Methods are currently being devised with the objective of circumventing SASP. As a case in point, maintaining a substantial hemin concentration is being considered as a potential solution [20].

Thus, cancer requires a specific redox state for optimal development. This state appears to be an appropriate level of oxidative stress, which is achieved by antioxidant regulation [6,14]. Furthermore, the specific redox state necessary for cancer development appears to vary depending on the type and stage of the disease [9]. It is our contention that this topic remains still poorly understood, and it may prove crucial in developing new anti-cancer strategies, including those for prevention, diagnosis, and treatment. Therefore, the paper’s main goal was to find the effect of neoplasm disease on the redox balance of the human organism, including reactions related to antioxidants. Consequently, by a comprehensive account of the existing knowledge on this subject, this review article may prove to be a valuable guide for more effective strategies for combating cancer, which is the secondary objective of the article.

## 2. Methods

The subject literature was collected using the most important bibliographic databases in medical biology, biochemistry, and related sciences. PubMed, Embase, EBSCO, Scopus, and Web of Science bibliographic databases were searched using medical subject headings (MeSH) combinations and free text terms. The most important keywords typed into the search engines were “oxidative stress”, “antioxidants”, “reactive oxygen species”, “reactive nitrogen species”, “free radicals”, “free radical scavengers”, “neoplasms”, “therapeutics”, and “diagnosis”. Logical operators (AND/OR) were used to intersect or combine relevant terms. The literature searched was mainly the most recent, from the last 10–15 years. Older references were also used when warranted (e.g., source of information, historical facts, valuable original research). The literature search also included the widely available Google Scholar search engine.

## 3. The Oxidant–Antioxidant Equilibrium

Redox balance is also referred to as oxidant–antioxidant equilibrium, since it encompasses both oxidation and reduction reactions, which entail the donation and acceptance of electron(s), respectively [21]. In essence, antioxidants serve as reductants within an organism, whereas ROS and RNS function as oxidants. In the reaction of the oxidant with the antioxidant, the former undergoes reduction, while the latter undergoes oxidation (Figure 1). Therefore, antioxidant recovery mechanisms using other sources of antioxidant capacity are required [22]. The common unit of antioxidant capacity, or universal reductant, is nicotinamide adenine dinucleotide phosphate (NADPH) [23,24]. The cytosolic fraction of this dinucleotide is formed in the oxidative phase of the pentose-phosphate pathway via NADP-dependent dehydrogenases (6-phosphate dehydrogenase and 6-phosphogluconate dehydrogenase) during glycolysis (parallel processes). Moreover, enzymes linked to the Krebs cycle (isocitrate dehydrogenase, malic enzyme, and glutamate dehydrogenase) and enzymes of folate metabolism have been shown to be a significant source of NADPH in both the cytosol and mitochondria. These enzymes are of particular importance during periods of increased demand for dinucleotide (e.g., anabolic processes related to cell division) [25]. NADPH functions as an electron and proton (hydrogen) donor and is an integral part of every redox cycle, which is a series of oxidation–reduction reactions initiated by the interaction between the primary antioxidant and the primary oxidant, with regeneration of the antioxidant at the end [26,27].

### 3.1. Reactive Oxygen Species

ROS can be defined as oxygen atoms and oxygen-containing molecules that possess relatively high reactivity within biological systems. ROS that contain a nitrogen atom are designated as RNS. An illustrative example is peroxynitrite (ONOO^−^), which exhibits pronounced oxidizing properties, and nitric oxide (NO˙), a vasodilator with relatively weak oxidizing properties. Both ROS and RNS can be either free radical or non-radical in nature. The defining characteristic of free radicals is the presence of unpaired electrons in the valence shell (e.g., superoxide anion radical, O_2_^−^, hydroxyl radical, ˙OH, and NO˙). In contrast, non-radical particles lack this unpaired electron configuration but retain their high reactivity (e.g., singlet oxygen, ^1^O_2_, ozone, O_3_, hydrogen peroxide, H_2_O_2_, and ONOO^−^) [28,29].

The primary exogenous sources of ROS within the human body include ultraviolet radiation, ionizing radiation (α, β, γ, X), xenobiotics (e.g., pharmaceuticals, dioxins, polycyclic aromatic hydrocarbons such as benzopyrene, alcohol consumption, smoking), and a diet rich in simple sugars and hydrogenated fats (*trans* form) [7,22] (Figure 2). The primary endogenous source of ROS is some oxidoreductases, which are classified as the first group of enzymes in the commonly utilized enzyme catalog (enzyme catalog 1, EC 1). Many of these enzymes are a source of free radicals, O_2_˙^−^ and H_2_O_2_ [8,22]. Furthermore, reactions involving transition metals (e.g., Fe, Cu, Ni, Mn, Cr, Hg, Mo), collectively termed the Fenton reaction, are also an important source of free radicals, basically ˙OH (H_2_O_2_ + Fe^2+^ → ˙OH + OH^−^ + Fe^3+^) [30,31]. However, as to where free radicals are formed, the main source of them is the respiratory chain in the mitochondria, where atmospheric oxygen (O_2_) undergoes incomplete reduction by one electron. The result is the formation of O_2_˙^−^ in the amount of approximately 2% of atmospheric oxygen that is absorbed by the body [28,32]. This radical serves as a source of additional ROS in subsequent redox reactions that are catalyzed by enzymes. Superoxide dismutase (SOD) converts it to H_2_O_2_, which exhibits limited oxidative properties but demonstrates effective passive penetration through lipid membranes. Consequently, H_2_O_2_ represents a significant source of subsequent ROS and RNS at sites distant from the mitochondrion. In cells of the immune system, primarily macrophages and neutrophils, myeloperoxidase (MPO) catalyzes the enzymatic combination of chlorine with hydrogen peroxide, resulting in the formation of hypochlorous acid (HOCl). These cells also possess a particular enzyme, inducible nitric oxide synthase (iNOS), which produces NO˙, while nitric oxide forms peroxynitrite when combined with O_2_˙^−^. As previously stated, ONOO^−^ is a highly potent oxidant, in part due to its ability to spontaneously decompose into nitrogen dioxide and ˙OH, the most reactive form among ROS. The entire rapid sequence of free-radical reactions, which ultimately results in an “oxygen burst”, is initiated in phagocytes by NADPH oxidase (NOX), which produces O_2_˙^−^ [32,33]. ONOO^−^ can also be formed in vascular endothelium in the same reaction, albeit with endothelial-specific nitric oxide synthase (eNOS) serving as the source of NO˙ in this instance [22] (Figure 2).

The endogenous production of ROS is intensified in conditions of increased exercise intensity, mainly due to augmented cellular respiration [34]. As mentioned in the introduction, oxidative stress and the associated inflammation represent a significant contributor to tumorigenesis [3]. Furthermore, an elevated prevalence of neoplastic disease has been observed in the absence of regular physical activity [35]. This phenomenon may be explained by enhanced ROS production under physical exercise, which, according to the hormesis theory [36], increases antioxidant capacity [37].

ROS, including RNS, are implicated in the regulation of numerous physiological processes. They are needed for communication, growth, and differentiation [38]. As mentioned, ROS are also utilized as weapons in the fight against pathogens [32]. In the majority of cases, they are an intracellular tool for the regulation of cell activity. A series of mutual oxidation and reduction reactions are employed by the cell in transmitting signals (e.g., cysteine residues in the active sites of tyrosine phosphatases are susceptible to oxidative inhibition induced by H_2_O_2_) [38]. However, when their concentration exceeds the antioxidant capacity, they become toxic to the body. This is known as oxidative stress, which leads to molecular damage in the cell (oxidative damage to proteins, nucleic acids, lipids, and sugars). The damage results in compounds that are often toxic or simply non-functional. If the oxidative damage is not repaired, at least inflammation or necrosis occurs, depending on the concentration of ROS. Oxidative stress can also initiate apoptosis [28,33].

### 3.2. Antioxidant Capacity

Given the multitude of sources of ROS and their relatively straightforward formation within the body, redox imbalance is predominantly associated with oxidative stress. However, this does not occur if the body’s antioxidant defense system functions efficiently. This means that redundant ROS are scavenged by antioxidants, whereby the concentration of oxidants is kept in check within a spectrum of physiological, tissue-specific concentrations (the oxidant–antioxidant balance). Oxidative stress and increased cancer incidence are primarily observed in instances where there is a considerable influence of robust exogenous sources of ROS (Figure 2) [6,22] and in cases where there are genetic defects (mutations of DNA repair genes), e.g., Cockayne syndrome, Bloom syndrome, and Louis-Bar syndrome [39].

The antioxidant defense system can be divided into three levels (Figure 3). The initial line of defense comprises elements that impede the progression of redox reactions, effectively neutralizing primary free radicals (**the prevention**). These are mainly antioxidant enzymes that scavenge superoxide anion radical and hydrogen peroxide. The aforementioned SOD (EC 1.15.1.1), as well as catalase (CAT; EC 1.11.1.6), glutathione peroxidase (GPX; EC 1.11.1.9), and peroxiredoxin (PRX; EC 1.11.1.15), are the most important in this regard [40]. The roles of CAT, GPX, and PRX are to facilitate the removal of H_2_O_2_ produced by SOD activity [41]. At relatively high concentrations of H_2_O_2_, CAT is the primary enzyme responsible for its removal. In contrast to GPX and PRX, it is not feasible to saturate catalase with hydrogen peroxide. Therefore, its activity is directly proportional to the H_2_O_2_ concentration [42]. CAT is also tightly bound with an NADPH molecule, which likely protects the enzyme from inactivation, as H_2_O_2_ is sufficient for the enzyme’s redox reactions [43]. At decreased concentrations of H_2_O_2_, GPX is also involved in the scavenging process (catalase activity). However, GPX activity primarily entails the reduction in hydroperoxides through a process that involves reduced glutathione (GSH), glutathione reductase (GR, EC 1.8.1.7), and NADPH [44]. CAT, on the other hand, may also oxidize alcohols, which is called peroxidase activity of catalase, although the enzyme is referred to as a monofunctional enzyme (H_2_O_2_ decomposition) [42]. Finally, hydrogen peroxide and other hydroperoxides can also be neutralized by PRX, which utilizes thiol groups from cysteine residues. The activity of the enzyme is dependent on thioredoxin (TRX, EC 1.8.4.16) and sulfiredoxin (SRX, EC 1.8.98.2) [40,45]. TRX, in turn, depends on thioredoxin reductase (TR, EC 1.8.1.9) and NADPH. Together, they constitute the most essential disulfide reductase system (TRX system) [46]. This line of defense also encompasses other enzymes (e.g., ceruloplasmin/ferroxidase, CP/FOX, EC 1.16.3.1, glutaredoxin, GRX, EC 1.20.4.1) and non-enzymatic proteins that chelate transition metals (e.g., albumin, ferritin, and transferrin) [28].

The second line of defense comprises mainly low-molecular-weight compounds that exhibit reducing properties when in contact with ROS and are known as free radical scavengers. While the first level of antioxidant defense is comprised solely of high-molecular-weight endogenous antioxidants (proteins), the second level encompasses both low-molecular-weight endogenous compounds (e.g., glutathione, bilirubin, uric acid, and carnitine) and low-molecular-weight exogenous compounds (e.g., vitamins A, C, and E and flavonoids). These substances interact with ROS, impeding their capacity to initiate redox reactions, thereby disrupting the process of free radical-mediated cell destruction [22]. The second line of antioxidant defense is therefore distinguished by its interventionist nature (**the intervention**) [22]. GRX/GSH and TRX/TR can also be considered the third line of defense, as they are involved in the removal of disulfide bridges. Additionally, GSH is regarded as the main intracellular low-molecular-weight antioxidant due to its numerous antioxidant functions and considerable abundance [40,47].

The third line of defense is **the repair** of oxidative damage, which concerns mainly proteins. Proteins that have sustained severe oxidative damage are typically degraded in proteasomes (ATP- and ubiquitin-independent mechanism). Nevertheless, it is feasible to eliminate disulfide bridges (-S-S-) that have been introduced into a protein as a result of oxidative stress [28]. This elimination, namely the reduction of the bridges to free thiol (-SH) moieties, is performed by GRX or TRX, which are independent in this process due to the requisite number of -SH residues (two thiol moieties of cysteine in the active center). GSH and TR are then utilized to recover functional (reduced) GRX and TRX. The resulting GSSG is reduced to two GSH by GR [47]. Both GR and TR involve NADPH and their own reductases in the final restoration of SH residues [25,40]. The third line of antioxidant defense also comprises a number of highly specialized enzymes. However, in the context of this study, methionine sulfoxide reductase appears to be a more important example. This enzyme reduces oxidized methionine residues (MSR) in proteins and is known to have ROS scavenger properties. The literature reports a significant correlation between decreased concentration of MSR A and tumor progression [48,49].

## 4. The Redox Disturbances in Cancer

The imbalance of the oxidant–antioxidant balance in cancer involves excessive concentrations of ROS and oxidative stress. One of the original papers in the field of the oxidant–antioxidant balance in cancer examined a cohort of 29 individuals with colorectal cancer (adenocarcinoma) [50]. Cancer tissue was the study material (colonic mucosal samples extracted during the surgical resection of the tumor). In this way, the tumor microenvironment was studied under conditions of tumor budding and inflammatory infiltration. The activities of NOX (*p* < 0.01), xanthine oxidase (XO, *p* = 0.01), SOD (*p* < 0.0001), and CAT (*p* < 0.0001), as well as the total antioxidant capacity (TAC, *p* < 0.01), were found to be significantly elevated in cancerous tissue in comparison to normal colonic mucosa (histologically examined healthy tissue from the patients). Moreover, the levels of oxidative damage products were elevated in tumorigenic colon tissue, including advanced glycation end products (AGEs; *p* < 0.01), advanced protein oxidation products (AOPPs; *p* < 0.001), MDA (*p* < 0.001), and 8-OHdG (*p* < 0.0001). In addition, it was observed that the activities of CAT and XO in patients increased in correlation with the intensity of inflammatory infiltration (*p* < 0.05). Oxidative stress index (OSI) (*p* < 0.05) and MDA (*p* < 0.01) values were observed to be increased in the patients exhibiting a higher degree of tumor budding. However, no statistically significant difference was observed in the level of OSI between cancerous and normal tissue. The outcomes of this study confirm that colorectal cancer is associated with enzymatic and non-enzymatic redox imbalances and an increased degree of oxidative damage to proteins, lipids, and DNA. Concurrently, the observed changes suggest that oxidative stress within the tumor was managed by antioxidant enzymes (SOD, CAT), as evidenced by the lack of variation in OSI compared to normal tissue. The cancer had reached full development but had not yet metastasized [50]. Comparable redox imbalances were found in the plasma and serum of peripheral blood in patients with colorectal adenocarcinoma 24 h prior to surgery. Patients exhibited higher levels of oxidative stress and antioxidant capacity in comparison to the healthy controls. 24 h following the surgical excision of the tumor, a statistically significant decrease in oxidative stress markers and an increase in antioxidant markers were observed [51]. Oxidative stress is commonly observed in cancer. In a similar investigation, the authors revealed that peripheral blood redox markers indicated substantial oxidative stress in patients with colorectal adenocarcinoma [52]. Plasma and serum oxidative stress in colorectal cancer was also found by other researchers [53,54], demonstrating a positive relationship between cancer stage and MDA concentration [55], as well as identifying a new marker of oxidative stress in the form of ischemia-modified albumin (IMA) [56]. Increased levels of oxidative stress parameters were also observed in the urine of colorectal cancer patients [57]. Analogous findings were noted in patients with gastric cancer regarding blood assays [58,59], also indicating IMA as a marker of oxidative stress [60] (Table 1). Increased IMA concentration was also shown in children with soft tissue sarcomas and neuroblastoma [61].

In a further study, the authors evaluated the oxidant–antioxidant equilibrium and additionally assessed lysosomal and antiprotease activities in peripheral blood samples from 15 women with ovarian cancer in an advanced stage of the disease (tumor recurrence and multiorgan metastases) [63]. Plasma vitamin E concentration and serum 8-iso-prostaglandin F2α (8-iso-PGF2α) concentration (a marker of oxidative stress-induced lipid peroxidation), as well as arylsulfatase activity, were observed to be lower in the ovarian cancer patients compared to the control group (*p* = 0.006, *p* = 0.03, *p* = 0.001, respectively). While the activity of cathepsin D in serum was found to be lower in the control group compared to the patients (*p* = 0.04), no statistically significant differences were observed between the patients and the controls with respect to the remaining parameters [63]. The activities of SOD, GPX, and CAT were examined in erythrocytes, along with the concentrations of conjugated dienes (CD) and thiobarbituric acid reactive substances (TBARS). Additionally, CD and TBARS concentrations were determined in plasma, as were vitamin A concentrations. TAC and the concentrations of 4-hydroxynonenal adducts (4-HNE) were also examined in serum, along with the activities of acid phosphatase (a lysosomal enzyme) and α1-antitrypsin (an antiprotease factor). In light of these findings, the authors identified only mild anomalies in the oxidant–antioxidant and lysosomal systems. It seems probable that they observed an elevated oxidative potential accompanied by an increased antioxidant capacity in the affected women. However, oxidative stress resulting in perforation/destruction of lysosomal membranes and release of enzymes from lysosomes, which may also be accompanied by inflammation, cannot be clearly stated [63]. Nevertheless, other studies have unequivocally found oxidative stress in breast cancer patients, both in serum [64,65] and urine (only increased AOPPs) [57]. Disturbed redox equilibrium in favor of excessive ROS concentrations was also indicated in serum and plasma of patients with lung [66] and bladder [67,68] cancers, while, in patients with endometrial carcinoma, this was also associated with inflammation [69]. In turn, Nayyar et al. suggested serum oxidative stress also in oral squamous cell carcinoma [87]. In addition, Kundaktepe et al. posit that overproduction of ROS leads to oxidative stress in the breast and colon, which can potentially result in cancerous developments in these sites. The authors postulate that oxidative stress plays a crucial role in the development of solid malignancies [62].

### 4.1. NRF2 and HIF Signaling Pathways

ROS are integral to cell signaling at the fundamental atomic level [5]. For instance, the role of ROS in regulating protein activity at the post-transcriptional level has been demonstrated. As previously stated, an illustrative example is the inhibition of tyrosine phosphatases through the oxidation of their cysteine residues, which is induced by H_2_O_2_) [38]. In cancer cells, ROS condition proliferation and metastasis. They facilitate tumor progression, among other mechanisms, by enabling adaptation to hypoxia [6]. However, elevated levels of ROS result in increased damage to DNA and other essential biomolecules, which can impede cancer cell survival. Consequently, while ROS are overproduced by cancer cells to drive hyperactivation, their concentration must be carefully regulated to avoid excessive damage and cell death. This underscores the necessity for robust antioxidant defenses within the tumor microenvironment [88]. During the promotion phase, the survival of tumor cells at elevated concentrations of ROS is contingent upon the activation of antioxidant transcription factors or the augmentation of NADPH concentration through the pentose phosphate pathway (PPP). During the progression and metastasis stages, the tumor undergoes adaptations to cope with oxidative stress, which include the elevation of NADPH concentration through the activation of 5’AMP-activated protein kinase (AMPK) and the reductive metabolism of glutamine and folic acid (Figure 4) [89].

The initial response of a cell to oxidative stress is the activation of the transcription factor called nuclear factor erythroid 2-related factor 2 (NRF2), while the subsequent response is dependent on transcription factors NF-κB and activator protein-1 (AP-1). The result is not only an increase in antioxidant capacity but also the initiation of an inflammatory response. When the level of oxidative stress is excessive, a third level of response is initiated, namely apoptosis (Figure 4). This occurs when the concentration of ROS exceeds the body’s capacity to defend itself and adapt. In the event of exceedingly severe oxidative stress, necrosis may also ensue [90]. In a growing tumor, hypoxia is an equally important and common phenomenon that occurs when the rate of development surpasses the capacity for new vascular network formation. A deficiency in oxygen results in the activation of responses dependent on hypoxia-inducible factor (HIF) [91].

The principal signaling pathway that determines the redox state of a tumor is NRF2 signaling [90,91]. The effect of NRF2 activity is to manage the transcription of more than 200 genes [92]. It functions as a transcription factor that regulates the HIF pathway as well. The binding of NRF2 to the antioxidant response element (ARE) can induce HIF-1α protein expression due to the proximity of the ARE and the HIF1A gene promoter [93]. This interaction has been observed in breast and bladder cancers [94] and cisplatin-resistant liver cancer cells [95]. Moreover, it has been demonstrated that NRF2 induces the expression of TRX1 in non-small cell lung cancer [96] and NADPH quinone oxidoreductase 1 (NQO1) in human colorectal and breast cancer cell lines [97], which have been shown to increase HIF-1α levels by prolonging its lifespan (stabilization). Thus, NRF2 can also increase the concentration of HIF-1α by safeguarding it from degradation in proteasomes. This is also linked to the fact that the binding of NRF2 to the oxygen-dependent domain in HIF-1α protects HIF-1α against ubiquitination and subsequent proteasomal degradation. ROS can also stabilize HIF-1α since they can inactivate the prolyl hydroxylase domain protein (PHD) that catalyzes the mentioned HIF-1α degradation [97,98]. Carbon monoxide (CO) produced by heme oxygenase-1 (HO-1) also plays a role in this process, and oxidative stress-induced NRF2 activation directly turns on HO-1 transcription [91]. It has been demonstrated that NRF2/TRX1 and HIF-1α levels increased in adenocarcinoma cells due to oxidative stress induced by intermittent hypoxia in the NOX1-dependent pathway [99], and CO increased HIF-1α concentration through translational stimulation and stabilization of this protein in human umbilical vein endothelial cells and primary human brain astrocytes [100]. Inhibition of NRF2 activity, in turn, has been observed to decrease the concentration of HIF-1α and HIF-2α in the cytosol, which is mediated by specific small RNA molecules (miR-181c-5p and miR-181a-2-3p) [101,102]. Interestingly, there also is a negative regulatory interaction between NRF2 and HIF. A relatively high concentration of NRF2 can inhibit the activity of HIF-α by reducing the concentration of ROS [103,104]. In turn, HIF-1α can suppress the expression of NRF2 (and HO-1) by elevating the levels of BACH1, a transcription factor that exerts a repressive influence over NRF2 [105]. It should also be noted that hypoxia is accompanied by oxidative stress, which does not necessarily involve the activation of the NRF2 pathway. Meanwhile, NRF2 activation resulting from oxidative stress is always accompanied by the upregulation of the HIF pathway, as described in the aforementioned mechanisms [91].

### 4.2. The Antioxidant System as an Anti-Cancer Strategy

The blocking of the NRF2 and HIF pathways has emerged as a key area of interest within the field of cancer treatment. In particular, there has been interest in natural chemicals with potential anti-cancer properties that would not be too toxic to the body. An illustrative example is triptolide, a diterpenoid with alkylating agent properties, isolated from *Tripterygium wilfordii* [106], which inhibited the expression of NRF2 and HIF-1α in myeloid leukemia cells by blocking both pathways, thereby preventing the expression of their target genes. Triptolide demonstrated such properties when combined with doxorubicin or imatinib, which the cancer cells had previously demonstrated resistance to. Triptolide thus restored the therapeutic capacity of chemotherapeutics [107]. Moreover, the simultaneous use of triptolide and idarubicin synergistically enhanced the apoptosis of leukemia stem cells [108]. Similarly, brusatol, a quassinoid of the triterpenoid group extracted from *Brucea javanica* [109], was observed to inhibit HIF-1α activity by enhancing its degradation in the human colon cancer cell line HCT116 under hypoxic conditions. The ultimate effect was a reduction in glucose consumption by cancer cells and an inhibition of their growth as a consequence of the blocking of the HIF pathway (lack of expression of pathway target genes, among others: VEGF, GLUT1, HK2, and LDHA). The effect of brusatol was also associated with a decrease in mitochondrial ROS [110]. This study also clearly suggests the inhibitory impact of brusatol on NRF2. Another study proved it in a model of cancer stem cells (CSCs) under hypoxia, as was the regulatory role of NRF2 over the HIF pathway [111]. In this study, the inhibition of NRF2 by brusatol resulted in the blockade of HIF-2α accumulation, which subsequently led to a reduction in the expression of CSC markers and the inhibition of CSC properties, including spherical growth. Conversely, the overactivation of NRF2, whether genetically or chemically induced, was observed to enhance the accumulation of HIF-2α in response to chronic hypoxia, thereby promoting tumor migration [111]. An additional example is cardamonin, a natural chalcone (lipid group, flavonoid subgroup) derived from *Alpiniae katsumadai* [112], which inhibited the growth of breast cancer cells in both in vitro and in vivo models. The cancer cells underwent metabolic reprogramming, exhibiting reduced glucose uptake and diminished lactate production and export. Oxidative stress-induced apoptosis of cancer cells was also observed. Cardamonin was demonstrated to impede HIF-1α expression through the blockade of the mTOR/p70S6K pathway. As a result, oxidative phosphorylation in mitochondria and the production of ROS increased. Increased production of ROS also resulted from the attenuation of the NRF2-dependent ROS neutralization system [113].

Moreover, it is noteworthy to mention the correlation between the NRF2 and HIF pathways and ferroptosis in the context of tumorigenesis. Ferroptosis is a form of regulated cell death that is caused by oxidative stress resulting from the Fenton reaction. Cell death is initiated by the fragmentation of lipid membranes, which is caused by the accumulation of lipid hydroperoxides (Figure 4) [91]. GPX4 plays a pivotal role in the prevention of ferroptosis. It has been shown that hypoxia, the HIF pathway, can protect against ferroptosis. However, there are also reports of increased susceptibility of cancer cells to ferroptosis in a hypoxic state. In contrast, the significance of the NRF2 pathway in this issue is unequivocal—NRF2 promotes mechanisms that prevent ferroptosis [91]. This remains an underexplored issue worthy of further investigation. The link between the NRF2 pathway and cancer resistance to treatment is also worthy of further research. NRF2 can enhance the resistance of cancer cells to chemotherapeutic drugs via increasing antioxidant capacity [114]. This was found in many types of chemoresistant cancers [115], including, among others, cancers of the ovary [116], breast [117,118], head and neck [119], and gastrointestinal system [120]. However, targeting the NRF2 pathway may reverse the resistance, as mentioned in the case of triptolide, brusatol, and cardamonin. Isoorientin, which exhibits antioxidant, anti-cancer, and anti-inflammatory properties, acts in a similar way [121].

While the activation of the NRF2 pathway typically initiates the response to oxidative stress, recent reports have shown that different antioxidant system components can be practical targets of anti-cancer therapy. For example, recent studies have revealed the crucial involvement of TRX and GSH in the differentiation of effector and memory T cells [122,123] and the macrophage inflammatory response [124,125]. This is also the case with regard to B lymphocytes and their capacity to produce antibodies [126]. Activated immune cells engage the NF-κB and NLRP3 pathways, which facilitate the proliferation of cells (nucleotide biosynthesis) and the initiation of an inflammatory response [127]. Moreover, tumor cells employ TRX to facilitate their rapid growth while avoiding oxidative damage that could lead to their death (e.g., diffuse large B-cell lymphoma [128] and lung [129], colorectal [130], pancreatic [131], gastric [132], breast [133], and prostate cancers [134]). Accordingly, targeting the antioxidant mini-systems that recover TRX and GSH appears to be a promising anti-cancer strategy. The challenge, however, is selective targeting, i.e., targeting only cancer cells and only a specific redox system. This is because the objective is to achieve the greatest possible treatment effectiveness with the fewest possible side effects, which in turn requires careful consideration of the effects of the therapy on healthy tissues [135,136]. A reduction in the effectiveness of the TRX and/or GSH systems in immunocompetent cells may culminate in a diminished anti-tumor immune response [33]. Further studies thus seem warranted, for example, to determine the precise functions of the TRX pathway in distinct CD4+ T cell subpopulations, including TH1, TH2, TH17, regulatory T (Treg), or natural killer (NK) cells. Myeloid cells, including neutrophils, conventional dendritic cells (cDC1, cDC2), and inflammatory and alternatively activated macrophages, also play a significant role. Moreover, it would be beneficial to investigate the expression and function of the TRX system in exhausted T lymphocytes (Tex), as they are known to have reduced proliferative potential and accumulated ROS [137].

Negative regulation of antioxidant systems can enhance the efficacy of anti-cancer therapies that are based on chemotherapeutics and ionizing radiation. It is associated with the mitigation of adverse effects and an improvement in the patient’s quality of life. The next important issue in this context relates to antioxidant molecules, which have been identified as a highly effective form of alternative and complementary cancer therapy, offering therapeutic and preventive benefits. However, recent scientific evidence is emerging that challenges the efficacy or even the safety of this treatment [10]. Exogenous, low-molecular-weight antioxidants, which can be delivered to the tumor relatively easily, have become a widely utilized approach for this purpose. However, the intricate nature of the redox system presents a significant challenge in affecting redox reactions in cancer [90]. For example, high doses of antioxidants can be toxic, as these compounds can themselves undergo conversion to free radical forms in reactions with ROS [88]. Some studies indicated that antioxidant supplementation significantly increased cancer incidence rather than providing any protective effect or expected health benefits [138,139]. Basically, the free radical forms of low-molecular-weight antioxidants are relatively unreactive; however, in the absence of NADPH-based reducing agents (decreased antioxidant potential), they can become toxic [88]. In addition, the interplay between the redox and metabolic systems and their “high flexibility” (adaptability) indicate that targeted antioxidant therapy may result in unintended adaptations of the tumor, that is, changes that favor its survival. The heterogeneity of redox and metabolic processes within the tumor and peri-tumor tissues is also a contributing factor [140]. Therefore, the intricate subject of cancer antioxidant therapy must be more fully comprehended.

## 5. Antioxidants—Cancer Prevention and Treatment

### 5.1. Benefits

Antioxidants regulate the concentration of ROS, thereby modulating the immune response, activating suppressor genes, and exerting an inhibitory effect on oncogenes and angiogenesis. This applies to both endogenous and exogenous antioxidants [141,142]. A long-term study on a large cohort (*n* = 5141) demonstrated a favorable impact of a combination of antioxidant vitamins and minerals on prostate cancer incidence. However, this effect was observed exclusively in men with normal prostate-specific antigen (PSA) levels in venous blood plasma (<3 µg/L) prior to this study. In men with an elevated baseline PSA concentration, supplementation did not prevent an increase in cancer incidence. The supplementation regimen entailed the daily ingestion of a mixture comprising β-carotene (6 mg), α-tocopherol (30 mg), vitamin C (120 mg), selenium (100 μg), and zinc (20 mg) for 8 years. The reference point was men taking a placebo. Moreover, supplementation had no effect on either PSA or insulin-like growth factor (IGF) concentrations [70]. In a related study, Li et al. showed that supplementation with selenium, lycopene, α-tocopherol, and γ-tocopherol was associated with reduced risk of prostate cancer in patients with manganese SOD (MnSOD) polymorphism, which is a potential risk factor for the disease (567 patients, 764 controls) [71]. Among minerals, those that constitute the active component of oxidoreduction proteins are of particular significance. Specifically, these include selenium, which is a cofactor of GPX and TR; zinc and copper, which are components of SOD [10]; and iron, which is part of CAT [143]. A substantial body of evidence from in vitro (selenium [144,145], zinc [146], copper [147,148], and iron [149,150]) and in vivo studies (selenium [151], zinc [152,153], copper [154], and iron [155,156]) suggests that these minerals possess anti-tumor efficacy, including the ability to enhance the effects of commonly used chemotherapeutics (e.g., cisplatin [157,158]). Similarly, antioxidant vitamins (A [72], C [73,74], and E [75]) and plant polyphenols (e.g., quercetin [77] or α-lipoic acid [78]), particularly associated with regular and moderate physical activity [159,160], as well as endogenous compounds with antioxidant properties, such as bilirubin [79,80], coenzyme Q10 [76], and melatonin [81,82], have been linked to cancer prevention and treatment.

Of particular importance for antioxidant defense is the aforementioned triad of enzymes, namely SOD, CAT, and GPX. There is substantial evidence to suggest that reduced SOD activity plays a role in many oncogenic signaling processes. A reduction in copper–zinc SOD (CuZnSOD, SOD1) activity has been documented in the blood of individuals diagnosed with breast cancer [161] and in gastric adenocarcinoma tissue [162]. In both instances this was unrelated to SOD concentrations, as they were similar to those observed in healthy control subjects. Studies on the deficiency of SOD1 activity in mice have shown a tendency for the formation of nodules in the liver, including malignant tumors [163]. Furthermore, the expression of the extracellular form of CuZnSOD (SOD3) has been found to be diminished in numerous cancers, including those of the lung [164], prostate, pancreas, thyroid, and colon [165]. While MnSOD (SOD2) appears to be particularly important in pancreatic cancer, the enzyme’s activity is markedly reduced in both the cancer cells and the stroma tissue that surrounds the cancer [166]. A negative correlation between SOD2 activity and the rate of proliferation has been shown in cell lines of this cancer. The majority of patients with pancreatic cancer present with a mutation in the K-RAS oncogene, which alters the redox state of pancreatic cells to favor malignant proliferation. This is most likely related to increased NOX activity and thus overproduction of O_2_˙^−^ [166]. Additionally, a reduction in SOD2 activity has been documented in female breast cancer tissue [167]. Conversely, elevated MnSOD gene expression was associated with a reduction in the incidence of skin cancer in a murine model. However, a reduction in the gene expression did not result in an increased incidence of tumor formation, as both cell proliferation and apoptosis were enhanced [168]. There is substantial evidence supporting the efficacy of MnSOD gene therapy or synthesized SOD mimetics in the context of cancer treatment [169,170,171,172].

The increased expression of CAT led to a decline in the proliferation and migration rates of human MCF-7 breast cancer cells. Additionally, it augmented the sensitivity of these cells to anticancer treatment, including doxorubicin, cisplatin, and paclitaxel [173]. Similarly, in cultures of human lung cancer cells (A549). Treatment with cisplatin, 5-fluorouracil, and hydroxyurea, along with an increase in CAT activity, correlated with reduced tumor aggressiveness and proliferation [174]. Furthermore, Bracalante et al. revealed that CAT overexpression inhibits cell proliferation and reverses the amelanotic phenotype in A7 clones of human A375 amelanotic melanoma cells. The clones demonstrated a high proliferation rate under the elevated H_2_O_2_ concentration. It was also accompanied by enhanced antioxidant capacity. In conclusion, the augmented CAT activity led to the formation of less malignant tumor cells [175].

GPX activity has been reported to be increased in a number of tumorigenesis cases, including oral squamous cell carcinoma [176], lung [177], colorectal [178], breast [179], and brain [180] cancers. Nevertheless, as with SOD and CAT, elevated GPX expression also has the potential to promote anti-tumor effects. This was demonstrated in both in vitro and in vivo models using human pancreatic cancer cells. The expression of the GPX gene was enhanced by adenovirus. The inhibitory effect was found to be even more pronounced when MnSOD activity was similarly augmented [181]. These findings are similar to those reported in the other study. The authors used two forms of phospholipid hydroperoxide GPX, i.e., mitochondrial and nonmitochondrial. Both forms of GPX inhibited the development of pancreatic tumor cells in both in vitro (over 80%) and in vivo study models [182].

### 5.2. Controversies

The role of antioxidants in cancer prevention and treatment remains inconclusive. ROS can either initiate tumorigenesis and drive tumor growth or exert cytostatic (inhibition of growth) and/or cytotoxic (induction of apoptosis) effects on the tumor [142,183]. A number of studies have indicated a link between elevated antioxidant capacity in malignant tissues and enhanced cancerous growth, augmented malignancy, and heightened resistance to therapeutic intervention [14,184,185,186]. Gaya-Bover et al. evaluated the concentrations of MnSOD, CuZn-SOD, CAT, GPX, and other proteins in colorectal cancer cells and in non-cancerous tissue adjacent to the tumor via Western blot analysis [187]. The adjacent tissue exhibited higher concentrations of CAT and GPX in patients, whereas the tumor displayed elevated levels of MnSOD and acetylated MnSOD. Notably, these concentrations were more pronounced in stages II and III of tumor malignancy compared to I and pre-cancerous stages [187]. Other studies have also indicated the presence of unexpectedly elevated levels/activity of SOD in oncological patients [188,189]. The elevated expression and activity of MnSOD in tumor cells impairs the efficiency of oxidative phosphorylation. This, in turn, activates the Warburg effect, which involves increasing the contribution of glycolysis to energy production, thereby reducing the role of mitochondria [190]. It appears that SOD2 in the pre-cancerous state may act as a tumor suppressor, preventing the initiation of carcinogenesis. Conversely, once carcinogenesis is initiated, the dismutase may promote tumor progression and facilitate the transformation of cells into more malignant phenotypes, including increased proliferation and metastasis. However, the role of MnSOD in this process is dependent on the cancer type, the MnSOD polymorphism, and redox mini-systems that collaborate with MnSOD (GPX-GR/PRX-TRX-TR), as the existing data on this subject are inconclusive [191,192]. Cancer cells that are undergoing rapid proliferation encounter a state of hypoxia, which prompts a series of metabolic alterations. First and foremost, highly proliferative cancer cells depend on glycolysis to meet elevated energy demands [183]. The activation of HIF-1 results in the suppression of pyruvate dehydrogenase (PDH). This, in turn, leads to the opening of glucose transporters and an increase in glucose flux in the glycolytic cycle. Furthermore, HIF-1 modulates cell gene expressions, shifting from glycolysis to the production and utilization of lactates (the Warburg effect). Positive feedback has also been observed from fumarate and succinate, which may act on HIF-1 to induce a “pseudo-hypoxic” response. In addition, mutations in p53 attenuate TP53-induced glycolysis and apoptosis regulator (TIGAR), resulting in a lack of inhibition of the glucose-6-phosphate dehydrogenase (G6PD) activity. This, in turn, leads to an increase in metabolic flux through glycolysis and the PPP. These changes are inherently linked to elevated ROS production [183]. However, this also involves activating antioxidant transcription factors or the augmentation of NADPH concentration necessary for neutralizing intracellular oxidative stress, as previously discussed [89]. The G6PD activity constitutes the initial irreversible step of the PPP to generate NADPH [183].

A study conducted on human lung cancer cells and in the culture of mouse models of this cancer (induction of tumorigenesis by the B-RAF and K-RAS oncogenes) demonstrated that the enrichment of the diet and culture medium with antioxidants in the form of N-acetylcysteine and vitamin E resulted in a reduction in ROS levels, oxidative DNA damage, and p53 gene expression, which in turn led to tumor progression [83]. Furthermore, a study of 1134 breast cancer patients treated with chemotherapy (cyclophosphamide, doxorubicin, and paclitaxel) found that the use of antioxidant supplements (vitamins A, C, and E, carotenoids, and coenzyme Q10) both before and during treatment was associated with an increased risk of tumor recurrence and, to a lesser extent, death [84]. The use of iron supplements both before and during chemotherapy was also found to be significantly associated with an increased risk of cancer recurrence. The use of multivitamins, on the other hand, was not associated with either cancer recurrence or survival [84]. A further study verified the effect of daily supplementation with antioxidant vitamins in combination with minerals (120 mg vitamin C, 30 mg vitamin E, 6 mg β-carotene, 100 μg selenium, and 20 mg zinc) on the incidence of skin cancer in 7876 women (35–60 years) and 5141 men (45–60 years) (double-blind, randomized controlled trial) [85]. The women and men were randomly assigned to either the experimental group, which received the supplement, or the control group, which received a placebo. During the course of this study, which had a median follow-up period of 7.5 years, 157 cases of all types of skin cancer were reported, including 25 cases of melanoma. This study revealed that women who took the supplement were more likely to develop skin cancer, with a statistically significant increase in the incidence of melanoma. The administration of the supplement to the male subjects yielded no discernible effect [85]. A meta-analysis of large clinical trials (totaling 109,394 subjects) demonstrated that β-carotene in high doses (20–30 mg per day), in conjunction with other vitamins administered in the form of a multivitamin supplement, was significantly associated with an increased risk of lung cancer in tobacco smokers [86]. RNS present in cigarette smoke have been evidenced to induce the formation of reactive forms of β-carotene, including 4-nitro-β-carotene, resulting in increased oxidative damage to lung cells [193].

## 6. Conclusions and Future Prospects

Malignant tumors are distinguished by a markedly elevated growth rate and propensity for dissemination throughout the body, particularly at the advanced stage of malignancy (metastasis). ROS are the fundamental element that initiates and promotes tumorigenesis. Their sufficiently high concentration is necessary for the progression of the tumorigenesis process and progression to the subsequent stages of malignancy. However, a high metabolic rate, chronic oxidative stress, and an elevated number of cell divisions result in the accumulation of errors and modifications, including oxidative damage, which ultimately leads to the apoptosis of cancer cells. However, tumors are capable of surviving under the most adverse conditions. For instance, they can initiate tumorigenesis in neighboring tissues when the primary tumor is undergoing apoptosis (SASP) or reduce the number of errors and improve their status by increasing antioxidant capacity. The second capacity may result in a decreased rate of oxidative phosphorylation and the Warburg effect as a consequence, that is, the increased involvement of glycolysis in energy production (typically increased glucose requirements of cancer). Adequate concentration and activity of antioxidants are required for a tumor to flourish, in addition to a sufficiently high oxidative potential. However, this issue is complex. Low-molecular-weight antioxidants (antioxidant supplements), including minerals with antioxidant significance (e.g., iron, selenium, and zinc), can promote carcinogenesis, which has also been observed during treatment with chemotherapeutics (this was not observed in the case of the use of multivitamins). However, as a rule, they show the opposite effect. Increased antioxidant capacity, especially increased antioxidant enzyme activity (e.g., SOD, CAT, and GPX), results in worse tumor status, although just as many of the exact opposite effects have been observed. Downregulation of antioxidant systems, particularly NRF2 inhibition, may yield better cancer status and its heightened resistance to treatment. It all depends on the redox state in a tumor, which varies between phases of tumorigenesis and types of cancer. The advancement of more precise and individualized treatment protocols represents a crucial avenue for further research in this field.

A potential solution to the challenges of cancer therapy is photodynamic therapy (PDT), which employs light-activated photosensitizing molecules that can regulate cellular redox balance by alterations in endogenous production of ROS. In contrast to radiotherapy, which employs ionizing radiation, PDT eradicates cancerous cells by inducing oxidative stress through non-ionizing radiation, with minimal impact on surrounding healthy tissues. The specificity of PDT is achieved by conjugating photosensitizers with molecules that are specifically targeted to the tumor. In conjunction with suitable adjuvants and immunostimulants, PDT is a promising approach to combating cancer [194]. A range of other personalized and targeted treatments are currently in use. These include the utilization of nanoparticles [194] and the concurrent implementation of diagnostics, a process referred to as “theranostics” [151].

## Figures and Tables

**Figure 1 biomedicines-13-01149-f001:**
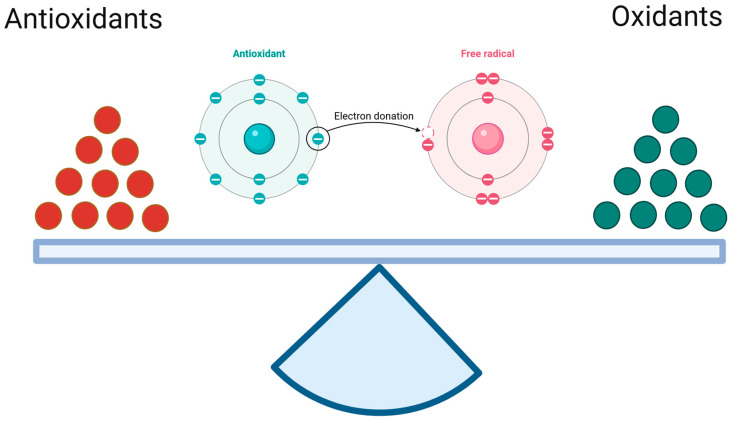
The oxidant–antioxidant equilibrium. It is a similar antioxidant/reduction potential to oxidation potential specific for the given tissue. (Created in BioRender. https://app.biorender.com/citation/68121ca6012f4aa59a973e05 (accessed on 30 April 2025)).

**Figure 2 biomedicines-13-01149-f002:**
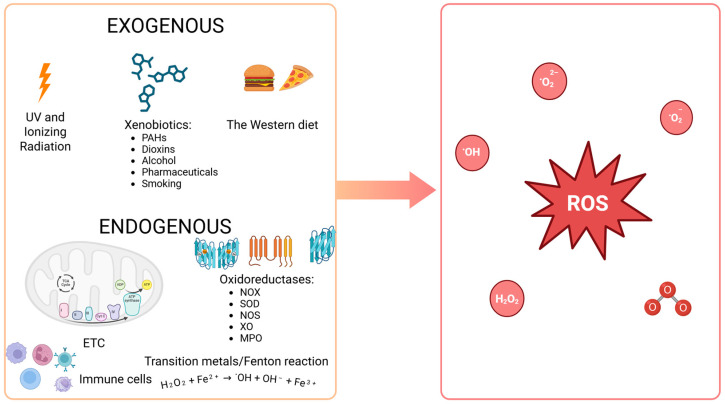
Sources of reactive oxygen species within the human body. PAHs: polycyclic aromatic hydrocarbons (e.g., benzopyrene); the Western diet: a diet high in fat and sugar; SOD: superoxide dismutase; NOX: NADPH oxidase; MPO: myeloperoxidase; NOS: nitric oxide synthase; XO: xanthine oxidase; ETC: electron transport chain in mitochondria; ROS: reactive oxygen species (e.g., ^1^O_2_: singlet oxygen; O_2_˙^−^: superoxide anion radical; H_2_O_2_: hydrogen peroxide; LOOH: lipid hydroperoxide; NO˙: nitric oxide; ONOO^−^: peroxynitrite; ˙OH: hydroxyl radical; HOCl: hypochlorous acid; O_3_: ozone). (Created in BioRender. https://BioRender.com/gqj3rqn (accessed on 30 April 2025)).

**Figure 3 biomedicines-13-01149-f003:**
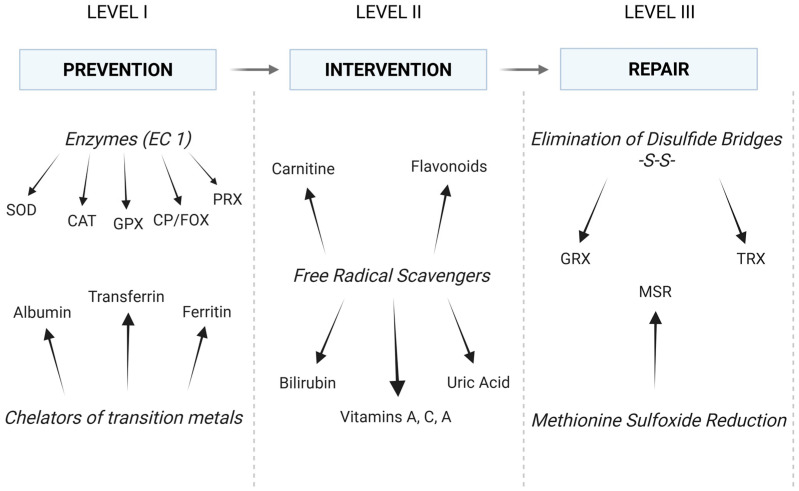
The antioxidant defense system: conventional division and main elements. In fact, elements of each level of defense collaborate to maintain a balance between all oxidation and reduction reactions (see text for details). EC: enzyme catalog; SOD: superoxide dismutase; CAT: catalase; CP/FOX: ceruloplasmin/ferroxidase; GPX: glutathione peroxidase; PRX: peroxiredoxin; GRX: glutaredoxin; TRX: thioredoxin; MSR: methionine sulfoxide reductase. (Created in BioRender. https://BioRender.com/c1szffk (accessed on 30 April 2025)).

**Figure 4 biomedicines-13-01149-f004:**
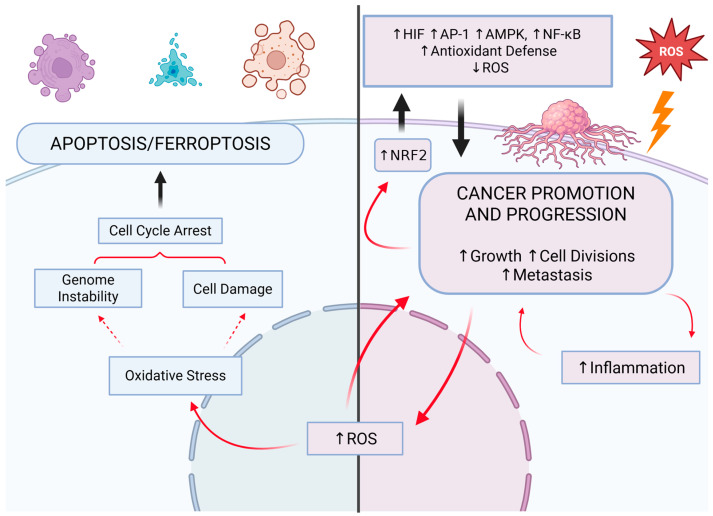
The relationship between cancer survival and development and the redox processes. The development of cancer is contingent upon the appropriate concentration of reactive oxygen species (ROS), which are regulated by antioxidants. Cancer cells may die as a result of excessive concentration of ROS in oxidative-stress-induced processes such as apoptosis, including ferroptosis. Excessive concentration of antioxidants results in a decrease in ROS concentration and inhibition of cancer growth. NRF2: nuclear factor erythroid 2-related factor 2; HIF: hypoxia-inducible factor; AP-1: activator protein-1; AMPK: 5′AMP-activated protein kinase; NF-κB: nuclear factor kappa B. Created in BioRender (Created in BioRender. https://BioRender.com/bo6c888 (accessed on 30 April 2025)).

**Table 1 biomedicines-13-01149-t001:** Redox changes in cancer and the impact of antioxidants on cancer.

Cancer Type/Study Material	Effect	References
Colon/colonic mucosa, peripheral blood	↑AC and ↑OS	[50,51]
Colon/peripheral blood	↑OS	[52,53,54,55,56,62]
Colon/urine	↑OS	[57]
Gastric/peripheral blood	↑OS	[58,59,60]
Ovarian/peripheral blood	Mild redox anomalies (tendencies toward ↑AC and ↑OS)	[63]
Breast/peripheral blood, urine	↑OS	[57,62,64,65]
Lung, bladder, endometrial, oral/peripheral blood	↑OS	[66,67,68,69]
**Caner type/antioxidant**	** *Positive* **	
Prostate/β-carotene, α-tocopherol, vitamin C, lycopene + minerals (Se, Zn)	Reducing cancer incidence	[70,71]
Neuroblastoma/vitamin A	Anticancer properties	[72]
Gastric, colon/vitamin C	Cancer prevention and anticancer properties	[73,74]
Prostate/vitamin E, coenzyme Q10	Anticancer properties	[75,76]
Breast/quercetin, α-lipoic acid	Anticancer properties	[77,78]
Colon/bilirubin, melatonin	Anticancer properties	[79,80,81]
Gastric/melatonin	Anticancer properties	[82]
	** *Negative* **	
Lung/N-acetylcysteine and vitamin E	Tumor progression	[83]
Breast/vitamins A, C and E, carotenoids, coenzyme Q10	Increased risk of tumor recurrence	[84]
Skin/vitamins C, and E, β-carotene + minerals (Se, Zn)	Increased incidence of cancer in women	[85]
Lung/β-carotene + vitamins (multivitamin supplement)	Increased risk of cancer in tobacco smokers	[86]

↑AC: antioxidant capacity; ↑OS: oxidative stress.

## Data Availability

No new data were created or analyzed in this study.

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
