# Peer review of "Redox Balance in Cancer in the Context of Tumor Prevention and Treatment"

_biomedicines, 2025, doi:10.3390/biomedicines13051149_

Round 1

Reviewer 1 Report

Comments and Suggestions for Authors

Manuscript ID: biomedicines-3607411 
“Redox Balance in Cancer in the Context of Tumor Prevention and Treatment” is a very good review. This review report is very interesting, well written and suitable for this journal.
Comments
1.    Introduction is too lengthy. Cut down the sentences from line 46 to 65. That paragraph is more descriptive. Remove antioxidant related paragraph (line 91 -98). explain this paragraph later with subheading like antioxidant regulation in cancer.
2.    In oxidant- antioxidant equilibrium you should include some schematic representation. 
3.    Fenton reaction, are also an important source of free radicals, basically Ë™OH (H2O2 + Fe2+ → Ë™OH + OH– + Fe3+). (line 173). You can represent in figure.
4.    For Figure 1, please use biorender platform and increase the quality of figure. 
5.    This phenomenon is 199 associated with exercise-induced ROS production, which, according to hormesis theory 200 [40], results in increased antioxidant capacity (line 199). Rewrite this sentence.
6.    Given the multitude of sources of ROS and RNS and their relatively straightforward 225 formation within the body, redox imbalance represents the most prevalent concern with 226 regard to oxidative stress. However, this does not occur if the body’s antioxidant defense 227 system is functioning efficiently, whereby the concentration of oxidants is then kept in 228 check, within a spectrum of physiological, tissue-specific concentrations (the oxidant–an-229 oxidant balance.(line 225). Sentence construction is not good. Rewrite: What is antioxidant capacity?
7.    You have to improve the figure 2 quality also. Now this is too simple.
8.    Thioredoxin group of antioxidants scavenging enzymes details also you should include in the antioxidant capacity paragraph.
9.    One of the original papers in the field of the oxidant–antioxidant balance in cancer 305 examined a cohort of 29 individuals with colorectal cancer (adenocarcinoma) (line 305). This sentence is not a good starting sentence. Please change according to the subheading.
10.    In the patients presenting with Conn syndrome, potassium or spironolac-357 tone, an aldosterone receptor blocker, was employed (line 357). Please check this meaning.
11. Paragraphs starting from lines 342 to 399 are too lengthy. Divide into 2 paragraphs.
12.    You can add more relationships among ROS, NRF,2 and HO-1.
13.    Inhibition of NRF2 activity, in turn, has been observed to increase 443 the concentrations of miR-181c-5p and miR-181a-2-3p, small RNA molecules that have 444 been demonstrated to negatively regulate the transcription of HIF-1α and HIF-2α, respec-445 tively, resulting in decreased concentration of these proteins in the cytosol. We don’t need this sentence here. Check it
14.    Figure 3, use better representation. Figure concept is ok, but too simple. 
15.    Prepare one table to discuss about cancer, redox change because of that particular cancer and antioxidant therapy 
16.    In benefits also you can add more invitro and in vivo data. 
17.    Also add post-translational signaling of ROS 
18.    Cancer metabolism and ROS also you can discuss in one sub heading. Because cancer cells are more active in aerobic glycolysis.
19.    Appreciated for including most relevant references

Author Response

Dear Reviewer,

Thank you very much for this peer review and for improving the manuscript. Please find below our responses. We have taken into account all comments. Other changes to the paper have been dictated by the comments of the other reviewer.

Best regards,

The authors

Comments and Suggestions for Authors

Manuscript ID: biomedicines-3607411 
“Redox Balance in Cancer in the Context of Tumor Prevention and Treatment” is a very good review. This review report is very interesting, well written and suitable for this journal.
Comments
1.    Introduction is too lengthy. Cut down the sentences from line 46 to 65. That paragraph is more descriptive. Remove antioxidant related paragraph (line 91 -98). explain this paragraph later with subheading like antioxidant regulation in cancer.

Authors’ response: corrected as suggested (the paragraph 91-98, now 93-100, moved to lines 596-600 and 606-608).

  1.    In oxidant- antioxidant equilibrium you should include some schematic representation.

Authors’ response: new figure has been created (Figure 1).

  1.    Fenton reaction, are also an important source of free radicals, basically Ë™OH (H2O2 + Fe2+ → Ë™OH + OH– + Fe3+). (line 173). You can represent in figure.

Authors’ response: corrected as suggested (the formula is represented in Figure 2).

  1.    For Figure 1, please use biorender platform and increase the quality of figure. 

Authors’ response: All the figures have been improved in the manuscript using BioRender.

  1.    This phenomenon is 199 associated with exercise-induced ROS production, which, according to hormesis theory 200 [40], results in increased antioxidant capacity (line 199). Rewrite this sentence.

Authors’ response: corrected (lines 226-229).

  1.    Given the multitude of sources of ROS and RNS and their relatively straightforward 225 formation within the body, redox imbalance represents the most prevalent concern with 226 regard to oxidative stress. However, this does not occur if the body’s antioxidant defense 227 system is functioning efficiently, whereby the concentration of oxidants is then kept in 228 check, within a spectrum of physiological, tissue-specific concentrations (the oxidant–an-229 oxidant balance.(line 225). Sentence construction is not good. Rewrite: What is antioxidant capacity?

Authors’ response: corrected (lines 257-263).

  1.    You have to improve the figure 2 quality also. Now this is too simple.

Authors’ response: corrected.

  1.    Thioredoxin group of antioxidants scavenging enzymes details also you should include in the antioxidant capacity paragraph.

Authors’ response: we have added this (lines 298-299), but only for clear signaling, since the TRX system is described further (lines 315-316 and 325-331).

  1.    One of the original papers in the field of the oxidant–antioxidant balance in cancer 305 examined a cohort of 29 individuals with colorectal cancer (adenocarcinoma) (line 305). This sentence is not a good starting sentence. Please change according to the subheading.

Authors’ response: corrected (lines 345-346).

  1.    In the patients presenting with Conn syndrome, potassium or spironolac-357 tone, an aldosterone receptor blocker, was employed (line 357). Please check this meaning.

Authors’ response: We have removed the part of that paragraph (lines 384-411, references 66, 67, 68) because, as the second reviewer rightly pointed out, the adrenal tumors mentioned in those papers are rarely malignant.

  1. Paragraphs starting from lines 342 to 399 are too lengthy. Divide into 2 paragraphs.

Authors’ response: We have kept that paragraph as one, after the previous correction.

  1.    You can add more relationships among ROS, NRF,2 and HO-1.

Authors’ response: added (lines 484-486), and, as you can see, we have revised whole that fragment (lines 480-492).

  1.    Inhibition of NRF2 activity, in turn, has been observed to increase 443 the concentrations of miR-181c-5p and miR-181a-2-3p, small RNA molecules that have 444 been demonstrated to negatively regulate the transcription of HIF-1α and HIF-2α, respec-445 tively, resulting in decreased concentration of these proteins in the cytosol. We don’t need this sentence here. Check it

Authors’ response: We decided to leave this sentence because it is an example of the last positive correlation between NRF2 and HIF. It is already followed by a mention of, also possible, negative relationships between these proteins. However, we simplified this sentence considerably (lines 492-497).

  1.    Figure 3, use better representation. Figure concept is ok, but too simple.

Authors’ response: corrected.

  1.    Prepare one table to discuss about cancer, redox change because of that particular cancer and antioxidant therapy

Authors’ response: Table 1 has been added.

  1.    In benefits also you can add more invitro and in vivo data.

Authors’ response: we have added one relevant reference (ref. 178, lines 689-693; the reference includes in vitro and in vivo study models).

  1.    Also add post-translational signaling of ROS

Authors’ response: in that part of the paper, we focused on suppression of oxidative stress by antioxidants, therefore we decided to add information about post-translational signaling of ROS in the previous chapter in the appropriate place (lines 444-447) (we have mentioned shortly about this in lines 245-248 (ref. 38).

  1.    Cancer metabolism and ROS also you can discuss in one sub heading. Because cancer cells are more active in aerobic glycolysis.

Authors’ response: added in relevant place (lines 722-737), referring to that what was mentioned in the lines 451-461.

  1.    Appreciated for including most relevant references

Authors’ response: as suggested; we have revised references during this revision throughout the paper.

Reviewer 2 Report

Comments and Suggestions for Authors

1.The introduction contains a lot of detail and theoretical data. At the end of the introduction, the journal's objectives are not clearly stated. It should include the first objective, the second objective, etc

2.The method section is a necessary condition in the manuscript, please add this section and describe the methodology adopted in carrying out your work.

3.The Oxidant–Antioxidant Equilibrium : This section is not included in the objectives set by the authors. Furthermore, its presence has no added value for the manuscript. It must be summarized while keeping both figures 1 and 2.

4.The Redox Disturbances in Cancer : Cushing's syndrome, conn, and pheochromocytoma are not cancers. Therefore, these findings cannot be included in the cancer conclusions.

Author Response

Dear Reviewer,

Thank you very much for this peer review and for improving the manuscript. Please find below our responses. We have taken into account all comments. Other changes to the paper have been dictated by the comments of the other reviewer.

Best regards,

The authors

Comments and Suggestions for Authors

1.The introduction contains a lot of detail and theoretical data. At the end of the introduction, the journal's objectives are not clearly stated. It should include the first objective, the second objective, etc

Authors’ response: corrected (lines 117-122)

2.The method section is a necessary condition in the manuscript, please add this section and describe the methodology adopted in carrying out your work.

Authors’ response: corrected (Section 2)

3.The Oxidant–Antioxidant Equilibrium : This section is not included in the objectives set by the authors. Furthermore, its presence has no added value for the manuscript. It must be summarized while keeping both figures 1 and 2.

Authors’ response: the main goal was to describe oxidoreductive processes in the body during the course of malignant neoplasm, so such a chapter seems justified to us. Corrected as suggested (we have shortened the section, removing fragments irrelevant to the message the figures bring).

  1. The Redox Disturbances in Cancer : Cushing's syndrome, conn, and pheochromocytoma are not cancers. Therefore, these findings cannot be included in the cancer conclusions.

Authors’ response: Thank you very much for this comment. We are not physicians, so the term “tumor” and the subject matter of the papers cited (oxidative stress; references 66,67,68) misled us. It is true that the incidence of malignant tumors increases with the size of the adrenal tumor, but most often such lesions are forms of benign adenomas. Malignant forms of adrenal tumors occur with a frequency of 1-12%. However, the authors did not carry out a diagnosis of cancer, so they did not specify what percentage of patients had a malignant tumor. Therefore, we have removed this section and the references from the paper.